# An *in-silico* human cell model reveals the influence of spatial organization on RNA splicing

**Zhaleh Ghaemi** [1,2]*, **Joseph R. Peterson** [1], **Martin Gruebele** [1,2,3], **Zaida Luthey-Schulten** [1,2,3]

**1** Department of Chemistry, University of Illinois at Urbana-Champaign, Urbana, Illinois, United States of America, **2** Beckman Institute for Advanced Science and Technology, University of Illinois at Urbana-Champaign, Urbana, Illinois, United States of America, **3** Department of Physics, University of Illinois at Urbana-Champaign, Urbana, Illinois, United States of America

* ghaemi@illinois.edu

**Data Availability Statement:** All relevant data are within the manuscript and its Supporting Information files. Additionally, the code we developed can be downloaded from Github

## Abstract

Spatial organization is a characteristic of all cells, achieved in eukaryotic cells by utilizing both membrane-bound and membrane-less organelles. One of the key processes in eukaryotes is RNA splicing, which readies mRNA for translation. This complex and highly dynamical chemical process involves assembly and disassembly of many molecules in multiple cellular compartments and their transport among compartments. Our goal is to model the effect of spatial organization of membrane-less organelles (specifically nuclear speckles) and of organelle heterogeneity on splicing particle biogenesis in mammalian cells. Based on multiple sources of complementary experimental data, we constructed a spatial model of a HeLa cell to capture intracellular crowding effects. We then developed chemical reaction networks to describe the formation of RNA splicing machinery complexes and splicing processes within nuclear speckles (specific type of non-membrane-bound organelles). We incorporated these networks into our spatially-resolved human cell model and performed stochastic simulations for up to 15 minutes of biological time, the longest thus far for a eukaryotic cell. We find that an increase (decrease) in the number of nuclear pore complexes increases (decreases) the number of assembled splicing particles; and that compartmentalization is critical for the yield of correctly-assembled particles. We also show that a slight increase of splicing particle localization into nuclear speckles leads to a disproportionate enhancement of mRNA splicing and a reduction in the noise of generated mRNA. Our model also predicts that the distance between genes and speckles has a considerable effect on the mRNA production rate, with genes located closer to speckles producing mRNA at higher levels, emphasizing the importance of genome organization around speckles. The HeLa cell model, including organelles and sub-compartments, provides a flexible foundation to study other cellular processes that are strongly modulated by spatiotemporal heterogeneity.

through: https://eukaryoticcellbuilder.github.io/HeLa_Builder/.

**Funding:** This work was supported by the NSF grants MCB-1244570 on the Evolution of Translation:From Molecules to Cells to Z.G. and Z. L.-S. and NSF Center for the Physics of Living Cells grant PHY-1430124 to M.G. and Z.L.-S.,the NSF Graduate Fellowship [grant DGE-1144245] to J.R. P., and the NIH P41-GM104601 to Z.L.-S. Z.L.-S held the Murchison-Mallory Chair and M.G. held the James R. Eiszner Chair while this work was carried out. Supercomputer time was provided by XStream-XSEDE [grant TG-MCA03S027]. The funders had no role in study design, data collection and analysis, decision to publish, or preparation of the manuscript.

**Competing interests:** The authors have declared that no competing interests exist.

## Author summary

The spliceosome is one of the most complex cellular machineries. It cuts and splices the RNA code in eukaryotic cells by dynamically assembling and disassembling. The components of spliceosome are formed in both the nucleus and the cytoplasm within the cell and primarily localized in nuclear membrane-less organelles. Therefore, a computational model of spliceosomal function must contain a spatial model of the entire cell. However, building such a model is a challenging task, mainly due to the lack of homogeneous experimental data and a suitable computational framework. Here, we overcome these challenges and present a spatially-resolved HeLa cell model, with nuclear, subnuclear, and extensive cytoplasmic structures. The three-dimensional model is supplemented by reaction-diffusion processes to shed light on the function of the spliceosome.

## Introduction

Cells use spatial organization to mediate their complex biochemical reaction networks. Although membranes have long been recognized as means to confine organelle-specific biomolecules, non-membrane-bound organelles are increasingly found to play crucial roles in cellular function [1]. Such membraneless organelles can be formed as liquid-liquid phase-separated regions, and are therefore also known as "liquid droplets" [2]. Cells have numerous such liquid droplets that form either in the cytoplasm or in the nucleus [3, 4]. Some types of droplets are known to be involved only in a specific cellular process. As a prime example, nuclear speckles, or interchromatin granules, are droplets formed in the nucleus that are thought to be primarily involved in pre-mRNA splicing [5].

The RNA splicing process that led to eukaryotic cell evolution enables cell complexity without massively increasing the gene count or genome size. Instead, the structure of genes evolved such that the coding regions (exons) are interrupted by non-coding regions (introns) [6]. Coding regions must then be ligated to form functional transcripts. A single gene can encode for a variety of functional proteins by a process called alternative splicing [7]. The spliceosome is the cellular machinery that binds to the intron/exon sites, removes the introns and joins the exon ends. It is a multi-megadalton complex consisting of five protein-RNA small nuclear ribonucleoprotein complexes (Uridine-rich snRNPs): U1 snRNP, U2 snRNP, U4 snRNP, U5 snRNP and U6 snRNP [8]. The biogenesis of splicing particles occurs in multiple steps spanning both the nucleus and cytoplasm, transports between these two compartments and terminates in the nucleus [6]. The mature splicing particles then localize in nuclear speckles and assemble on the pre-messenger RNA (pre-mRNA) transcripts in a coordinated and step-wise fashion. Upon completion of the splicing reaction, they disassemble [9], thus resulting in a highly dynamic machinery.

Analogous to the influence of molecular-level heterogeneity, organelle heterogeneity can lead to different cellular phenotypic behaviors [10]. However, the effect of variations in the organelles involved in the spliceosomal particles assembly (e.g., the number of nuclear pore complexes and the size of the nucleus), is yet to be investigated. There is yet another motivation for studies in this realm. It is well-recognized that the particle-assembly in most species, although not yeast, occurs in multiple compartments (the nucleus and cytoplasm) and sub-compartments [6]. What is missing in this context is a quantitative rationale for the shuttling of the precursors of the splicing particles between different compartments. Thirdly, whereas the basic utility of nuclear speckles in pre-mRNA splicing is appreciated [11], the influence of spatial localization on splicing activity and mRNA production is not quantitatively

understood. We anticipate that the localization of splicing components influences the efficiency of splicing.

To enable a comprehensive view of RNA splicing and splicing components formation, we developed a detailed spatial model of a human cell containing the most relevant organelles. As explained in the next sections, almost a hundred different biomolecules are involved in splicing processes. Some occur in relatively small numbers, necessitating a stochastic treatment of the processes leading up to splicing. In particular cell-to-cell heterogeneity in a population and its phenotypic behavior demand this approach.

Here, we construct an 18-$\mu m$ spatially-resolved model of a HeLa cell from a library of experimental data such as cryo-electron tomography [12], mass spectrometry [13], fluorescence microscopy and live-cell imaging [11, 14–16], and -omics data [17, 18]. We develop kinetic models to describe the reaction network of RNA splicing in accordance with the known biological events. The kinetic models were incorporated into our spatially-resolved eukaryotic cell model, complete with organelles, compartments and biomacromolecules. We then perform simulations using stochastic reaction-diffusion master equations (RDME) with our previously-developed cell simulation software, Lattice Microbes (LM) [19, 20] for up to 15 minutes of biological time. Interfacing our model with the LM [19, 20], uniquely enables us to study the kinetics of cellular activity on the relevant biological time scale up to hours. LM takes advantage of multiple-GPU processors, and therefore benchmarks much faster than similar softwares (e.g., Virtual Cell), as we have shown previously [21, 22]. Therefore, simulating 15 minutes of biological time on multiple-GPU processors, for nuclear processes and entire human cell, takes 20 minutes and 15 hours of walltime, respectively. The utilized GPU architecture which is also available to other research groups (through supercomputers and GPU cloud computing) and the details of the working principles of LM are explained in the Methods section.

The assumptions of constructing the spatial model of the human cell and the reaction schemes include: 1. Our study aims to investigate the RNA splicing process and the main organelles that are directly associated with this process including: the nucleus, nuclear speckles, Cajal bodies and parts of the cytoplasm (ER, mitochondria, Golgi apparatus). Therefore, our spatial model of the human cell mainly includes these organelles/ compartments. 2. Because the average protein half-life in human cells is $\sim$ 9 hours [23], a quasi-steady cell state is assumed over the 15 minutes of biological time that was simulated. Therefore, processes such as transcription and translation of the genes encoding the proteins involving in the RNA splicing process were not explicitly modeled. 3. We focused on building the groundwork for an adaptable platform for construction and simulation of spatial models of human cells. Hence, the cellular organelles/compartments that have not yet been included in the current version of the model can be readily added through the provided Python code. We assume that any required reaction or spatial components will be added by the researchers of our community.

Our simulations, featuring 15 minutes of biological time of the first spatially-resolved human cell model, explore how cellular organization affects the efficiency of spliceosomal particle formation and pre-mRNA splicing. We find that changes in the number of nuclear pore complexes affect the number of assembled splicing particles, an effect that remains consistent for different nuclear sizes; and quantitatively, the formation of correctly-assembled splicing particles in multiple compartments is more efficient. We also show that even a slight increase in the relative localization of splicing particles in nuclear speckles simultaneously enhances mRNA production and reduces noise in the generated mRNA. We are able to rationalize that the properties of nuclear speckles evolved while subject to physical constraints, such as their

size and number. Finally, we predict that the organization of active genes around nuclear speckles affects mRNA production.

## Results

### Spatially-resolved model of a HeLa cell

We used a data-driven approach to construct a representative HeLa cell model that has not been available thus far. First, we gathered structural and -omics data from a variety of experimental studies [11–18, 24–27]. Then for organelles such as the endoplasmic reticulum (ER), we used protein composition percentages of HeLa cell organelles determined by mass spectrometry [13] to determine the fraction of the entire cell volume that is occupied by each organelle (see Methods for details). The mass percentages of organelles excluding water were approximated by volume percentages because of the fact that average proteins are composed of similar C/N/O/H atoms ratios with a density of $1.4 g/cm^3$.

Fig 1A and 1B show the overall HeLa cell model and a detailed view of the nuclear region. Assuming experimental growth conditions resulting in spherically-shaped cells [26, 28], a volume of 3000 $\mu m^3$ [24] leads to a cellular radius of 8.9 $\mu$m. The essential components of the cell for studying the RNA splicing processes are: the plasma membrane, the cytoplasm, the ER, mitochondria, the Golgi apparatus and the nucleus. The ER units were modeled by stochastic shapes using a cellular automata algorithm (see S1 Text and S1 Table for details and algorithm). ER units were distributed in the cytoplasm, spanning from the nuclear envelope to the plasma membrane, and were intertwined with other cytoplasmic organelles [29, 30]. The ER units made up ∼ 4.5% of the cell volume [13]. About 2000 rod-shaped mitochondria with dimensions of 0.9 $\mu$m × 0.5 $\mu$m were randomly placed throughout the cytoplasm, filling ∼ 11% of the total volume [13], additionally, a network-like mitochondria model was also constructed (see Methods section for details). A Golgi apparatus consisting of five stacked sheets, each with a thickness of 0.128 $\mu$m, was placed close to the nucleus [31].

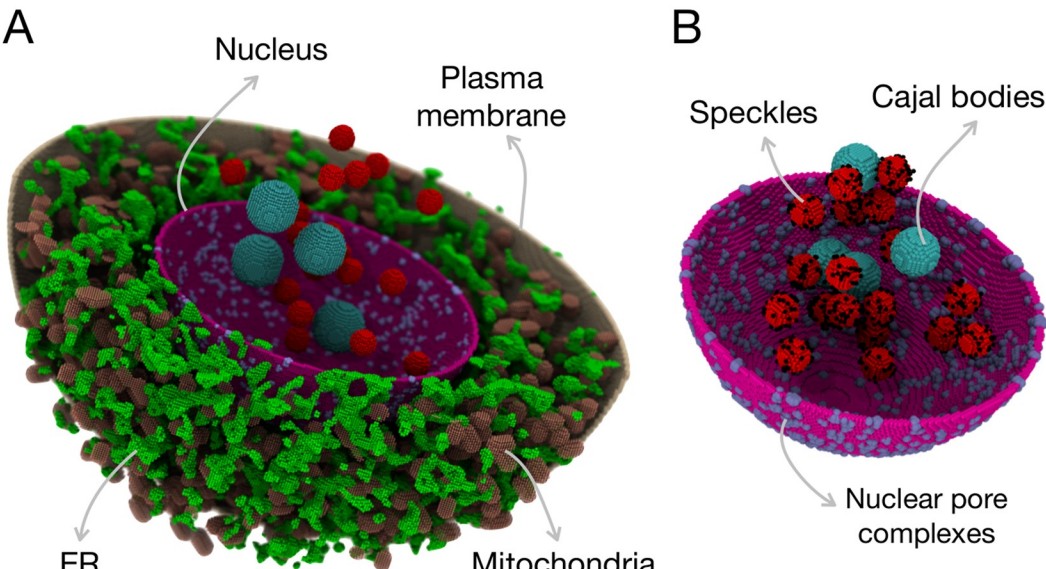

**Fig 1. A data-driven model for a 18-$\mu m$ HeLa cell: A) the cytoplasmic components are: ER, mitochondria and Golgi (not included in the figure for clarity); and B) a nucleus containing nuclear pore complexes, Cajal bodies and nuclear speckles.** The cytosol, nuclear pore complexes, nuclear speckles, and Cajal bodies are directly involved in RNA splicing processes, whereas ER, mitochondria and Golgi apparatus provide excluded volume effects.

**Table 1. The cellular components of the constructed HeLa model.**

| Component | Dimension ($\mu m$) | Number | Reference |
|---|---|---|---|
| HeLa cell | R = 8.9 | 1 | [24] |
| Nucleus | R = 3.7, 4.2, 4.7, 5.3 | 1 | [13, 25–27] |
| Nuclear pore complexes | R = 0.08 | 7/ $\mu m^2$ | [12, 14] |
| Mitochondria | $0.9 \times 0.5$ | 2000 | [15] |
| Nuclear speckles | R = 0.35 | 20 | [11] |
| Cajal bodies | R = 0.5 | 4 | [16] |
| ER | – | 4.5% cell volume | [13] |

The nucleus, which plays a critical role in our model, had a radius in the range of 3.7-5.3 $\mu$m [25–27]. It consists of nuclear pore complexes (NPCs) of 0.08 $\mu$m radii [12] at a density of 7 per $\mu m^2$ [14], 20 spherically-shaped nuclear speckles with 0.35 $\mu$m radii [11] and 4 Cajal bodies (sites of maturation of splicing particles) with 0.5 $\mu$m radii [16]. Actively transcribing genes (black dots in Fig 1B) were were randomly distributed within 0.02 $\mu$m of the edge of each nuclear speckle [32–34]. The nuclear components were chosen mainly among those that play a role in RNA splicing processes. Cellular components included in the *in-silico* model are listed in Table 1 along with their dimensions. The details of the construction of each organelle are provided in the Methods section.

## The kinetic model for spliceosome formation and action

We studied two processes: firstly, the formation of splicing particles U1snRNP and U2snRNP (U1 and U2 hereafter), which is a multi-compartmental process; secondly, spliceosomeal assembly, the splicing reaction itself and generation of mRNA transcripts. Together, these two series of events capture the whole process of splicing from machinery construction to functional transcript production. After the assembly of U1 and U2 in our model (the first process), the active genes are transcribed and pre-mRNA transcripts are spliced (the second process) according to the following reduced scheme for the spliceosome assembly:

1. An active-28 Kb gene is transcribed, without explicitly accounting for transcriptional machinery, and pre-mRNA transcripts are produced

2. U1 and U2 particles are formed and are present in the cell nucleus

3. Tri.U (U4/U6.U5) particles present in the nucleus; because of the complexity in the formation of these complexes we assumed they are pre-formed in our model [35].

4. The spliceosome assembles in a step-wise manner on pre-mRNA transcripts as described below

5. The splicing reaction occurs (described below) and an mRNA transcript is produced

6. The spliceosome disassembles after splicing, ready to assemble on another transcript

Below, we describe in detail the splicing particles' formation and splicing assembly and reaction.

**Formation of splicing particles.** A splicing particle consists of a uridine-rich small nuclear RNA (U snRNA) that is bound to a heptamer ring of Smith proteins (Sm), and a variable number of particle-specific proteins. The formation of splicing particles happens in multiple steps and compartments including the nucleus and the cytoplasm. To understand the

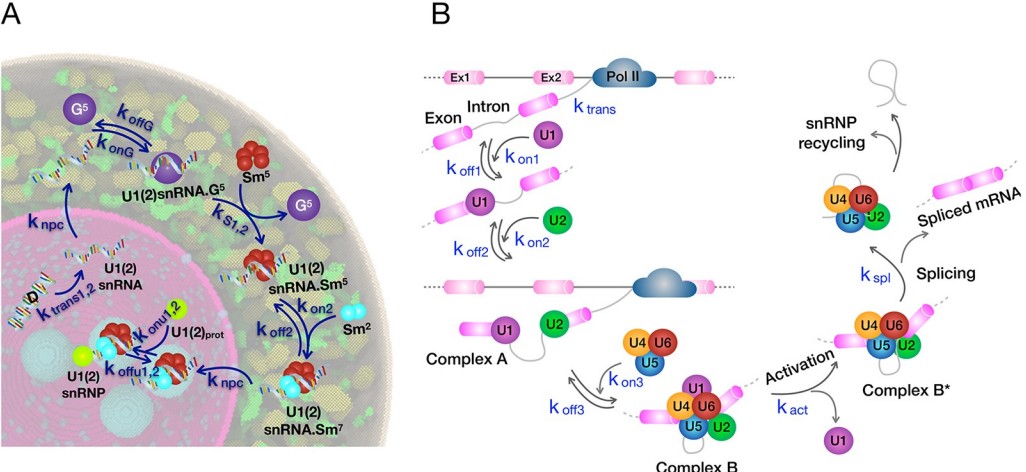

**Fig 2.** A) The reaction scheme describing the formation of U1 and U2 splicing particles mapped on a cross-section of our *in-silico* HeLa cell [38, 39]. The four spherically-shaped regions in cyan color are Cajal bodies. B) Splicing reactions as implemented in our simulations. The reactions together with their corresponding rate constants are presented in Tables 2 and 3. Abbreviations are: Pol II (RNA polymerase II), Ex1 and 2 (Exon 1 and 2).

effects of geometry on the assembly process, we developed a kinetic model to describe these processes and studied them in our spatially-resolved human cell model.

Fig 2A shows the steps associated with the formation of splicing particles. The reactions together with their associated rates are summarized in Table 2. Although not required for the scenarios studied here, our methodology has the flexibility to incorporate rate constants that are concentration dependent [36, 37].

Upon transcription, U1(2) snRNAs have to pass through nuclear pore complexes to reach the cytoplasm where, by a series of complex reactions, they bind to Sm proteins. Inspired by two studies [38, 39], we proposed the following mechanism for the cytoplasmic part of the process: the U1(2) snRNA transcript binds to Gemin 5 ($G^5$) which is part of the survival of motor neurons (SMN)-Gemin complex that mediates the Sm proteins assembly on snRNAs. The complex formed by this process then binds to a ring of five previously-assembled Sm proteins ($Sm^5$) through a process called RNP exchange suggested by Ref. [39]. This process facilitates the binding of Sm proteins to the snRNA transcript and the release of $G^5$. In the last step, the remaining Sm proteins ($Sm^2$) join the complex and the U1(2)snRNA.$Sm^7$ complex is formed. After the completion of the binding of Sm proteins to snRNA, the complex again passes through the nuclear pores and makes its way into the nucleus. In the nucleus, the U1(2)snRNA.$Sm^7_{nuc}$ complex localizes to the Cajal bodies [40] and binds to particle-specific proteins, U1(2)$_{prot}$, and as a result the mature splicing particle is formed. The diffusion coefficients for the species involved in the splicing particle formation reactions were mainly adopted from various experimental sources listed in S2 Table, and the concentration of species are reported in Methods.

**Assembly of the spliceosome and splicing reaction.**   The assembly process of the spliceosome machinery is entangled with a complex network of auxiliary and regulatory proteins that detect the splice site and vary the splice sites according to cellular cues by a process called alternative splicing [7]. To simplify this network, we assume that a particular splice site has been chosen according to the constitutive splicing process, and focus only on the assembly of the spliceosomal particles on that site and the subsequent splicing reaction.

Fig 2B depicts our model for the splicing reaction; and the details of the reactions and their associated rates are presented in Table 3. According to the conventional spliceoosome

**Table 2. Reactions describing U1 (u1snRNP) and U2 (U2snRNP) splicing particles formation together with their associated rates.** Abbreviations are: DNA(D), Gemin 5($G^5$), five already-assembled Sm proteins ($Sm^5$), the remaining Sm proteins ($Sm^2$), diffusion-limited (D.L.), model assumption (M), nucleus (N), NPC (P), Cajal bodies (J) and cytoplasm (C).

| Reaction | Rate | Units | Reference | Compartment |
|---|---|---|---|---|
| **In Nucleus** | | | | |
| $D \xrightarrow{k_{trans1}} D + U1snRNA_{nuc}$ | 0.285 | $s^{-1}$ | [62] | N |
| $D \xrightarrow{k_{trans2}} D + U2snRNA_{nuc}$ | 0.224 | $s^{-1}$ | [62] | N |
| **Nucleus to Cytoplasm** | | | | |
| $U1(2)snRNA_{nuc} \xrightarrow{k_{npc}} U1(2)snRNA_{cyt}$ | $2 \times 10^4$ | $s^{-1}$ | M | P |
| **Cytoplasmic Assembly** | | | | |
| $U1(2)snRNA_{cyt} + G^5 \xrightarrow{k_{onG}} U1(2)snRNA \cdot G^5$ | $1.02 \times 10^8$ | $M^{-1} s^{-1}$ | D.L. | C |
| $U1(2)snRNA \cdot G^5 \xrightarrow{k_{offG}} U1(2)snRNA_{cyt} + G^5$ | 3.05 | $s^{-1}$ | [63] | C |
| $U1snRNA \cdot G^5 + Sm^5 \xrightarrow{k_{S1}} U1snRNA \cdot Sm^5 + G^5$ | $5.9 \times 10^7$ | $M^{-1} s^{-1}$ | D.L. | C |
| $U2snRNA \cdot G^5 + Sm^5 \xrightarrow{k_{S2}} U2snRNA \cdot Sm^5 + G^5$ | $1.18 \times 10^7$ | $M^{-1} s^{-1}$ | D.L. | C |
| $U1(2)snRNA \cdot Sm^5 + Sm^2 \xrightarrow{k_{on2}} U1(2)snRNA \cdot Sm^7$ | $1.39 \times 10^8$ | $M^{-1} s^{-1}$ | D.L. | C |
| $U1(2)snRNA \cdot Sm^7 \xrightarrow{k_{off2}} U1(2)snRNA \cdot Sm^5 + Sm^2$ | 2.78 | $s^{-1}$ | [63] | C |
| **Cytoplasm to Nucleus** | | | | |
| $U1(2)snRNA \cdot Sm^7 \xrightarrow{k_{npc}} U1(2)snRNA \cdot Sm^7_{nuc}$ | $2 \times 10^4$ | $s^{-1}$ | M | P |
| **Nuclear Maturation** | | | | |
| $U1snRNA \cdot Sm^7_{nuc} + U1_{prot} \xrightarrow{k_{onu1}} U1snRNP$ | $1.22 \times 10^7$ | $M^{-1} s^{-1}$ | [64] | J-N |
| $U1snRNP \xrightarrow{k_{offu1}} U1snRNA \cdot Sm^7_{nuc} + U1_{prot}$ | $4.8 \times 10^{-4}$ | $s^{-1}$ | [64] | J-N |
| $U2snRNA \cdot Sm^7_{nuc} + U2_{prot} \xrightarrow{k_{onu2}} U2snRNP$ | $0.24 \times 10^7$ | $M^{-1} s^{-1}$ | [64] | J-N |
| $U2snRNP \xrightarrow{k_{offu2}} U2snRNA \cdot Sm^7_{nuc} + U2_{prot}$ | $4.8 \times 10^{-4}$ | $s^{-1}$ | [64] | J-N |

**Table 3. Spliceosome assembly and splicing reaction.** Abbreviations are: DNA (D), U1snRNP (U1), U2snRNP (U2), U4/U6· U5snRNP (tri· U), model assumption (M), diffusion-limited (D.L.), nuclear speckles (S) and nucleus (N).

| Reaction | Rate | Units | Reference | Compartment |
|---|---|---|---|---|
| $D \xrightarrow{k_{trans}} D + pre - mRNA$ | $4.7 \times 10^{-3}$ | $s^{-1}$ | [42] | S-N |
| $U1 + pre - mRNA \xrightarrow{k_{on1}} complexE$ | $4.66 \times 10^7$ | $M^{-1} s^{-1}$ | D.L. [16] | S-N |
| $complexE \xrightarrow{k_{off1}} U1 + pre - mRNA$ | 1.57 | $s^{-1}$ | [65] | S-N |
| $complexE + U2 \xrightarrow{k_{on2}} complexA$ | $8.8 \times 10^7$ | $M^{-1} s^{-1}$ | D.L. [16] | S-N |
| $complexA \xrightarrow{k_{off2}} complexE + U2$ | 0.062 | $s^{-1}$ | [65] | S-N |
| $complexA + tri \cdot U \xrightarrow{k_{on3}} complexB$ | $4.66 \times 10^7$ | $M^{-1} s^{-1}$ | D.L. [16] | S-N |
| $complexB \xrightarrow{k_{off3}} complexA + tri \cdot U$ | 1.55 | $s^{-1}$ | [65] | S-N |
| $complexB \xrightarrow{k_{act}} complexB^* + U1$ | $6 \times 10^4$ | $M^{-1} s^{-1}$ | M | S-N |
| $complexB^* \xrightarrow{k_{spl}} mRNA + tri \cdot U + U2$ | 0.067 | $s^{-1}$ | [50] | S-N |

assembly model [41], the U1 particle binds to the 5′ end of the exon ("complex E"), followed by binding of the U2 particle to the associate 3′ end to form "complex A". The tri.U (U4/U6. U5) then joins the complex forming "complex B". Subsequently, U1 leaves the complex, yielding the catalytically active "complex B*". The intron is then removed and the splicing particles are recycled for another round of assembly. The spliceosomal particle concentrations are reported in Methods and the diffusion coefficients are listed in S3 Table.

**Co-transcriptional splicing.** Splicing is known to be overwhelmingly co-transcriptional; as transcription occurs, the spliceosome assembles on the transcribed pre-mRNA and splicing reactions begin. In our model, an average gene consisting of 8 introns with an intron length of 3.4 Kbase (plus 137 base for each exon) is considered [42]. After the transcription of the first exon-intron-exon piece, the splicing reaction proceeds as discussed above. Simultaneously, another intron-exon pair is transcribed, continuing the spliced transcript. The cycle repeats until the end of the gene is reached.

## Organelle heterogeneity influences the formation of splicing particles

Cellular phenotypic variation arising from organelle heterogeneity is becoming a subject recognized as worthy of study [10]. We investigated how heterogeneity in NPC count and nuclear size affect the formation of splicing particles. All splicing particles, except U6, are complexes of uridine-rich small nuclear RNA bound to a heptameric ring of Sm proteins, along with specific proteins that bind to each splicing particles. Among the five splicing particles which are required for spliceosome function, we focus on the first two (U1 and U2) that start the spliceosome assembly process. As shown in Fig 2A and described above, these particles are formed in a multi-compartmental process [6]. Since splicing particles are assembled partly in the cytoplasm and partly in the nucleus, translocation through the NPCs is a crucial step. Live cell imaging has shown that the NPC count varies by as much as 10% [14], as does nuclear size [13, 25–27]. We hypothesized that these variations could influence the formation of splicing particles. We tested this hypothesis by varying NPC count and nuclear size, and examining the effect on the number of particles formed after 30 seconds of biological time, which was long enough to provide us with sufficient statistics. Performing RDME simulations of the entire HeLa cell model is computationally expensive but worthwhile in this case; it has been suggested that explicit representation of obstacles can clarify the extent of anomalous diffusion [43, 44]. Additionally, previous simulations of bacterial cells with spatial heterogeneity using LM [45], showed that the crowding within the cell provides an excluded volume effect and leads to anomalous subdiffusive behavior. Similarly, in our simulations (see simulation details in Methods), the cytoplasmic organelles (the ER, mitochondria and Golgi apparatus) can represent an upper bound for *in vivo* crowding effects. This effect is confirmed by a 28% increase in the production of U1 for a cell model without cytoplasmic organelles with respect to the full model. Therefore, we performed the splicing particles formation simulations involving reactions within the cytoplasm, with the entire HeLa cell geometry shown in Fig 1.

Likewise nuclear pores have an important effect on transport. Fig 3 demonstrates that increasing (decreasing) the number of NPCs by 20%, at a constant nuclear size, results in an increase (decrease) in the number of mature U1 and U2 splicing particles. The computed effect is consistent across the tested nuclear radii with a range from 3.7 to 5.3 $\mu$m [13, 25–27]. This suggests there are trafficking bottlenecks imposed by the nuclear pore count. We found that the number of splicing particles formed does not change significantly with nuclear size, as long as the number of NPCs scales with size. Longer diffusion times in a larger nucleus are compensated by shorter translocation times required when there are more NPCs. This trend is consistent with our observation that for a constant nuclear radius, decreasing the number of NPCs,

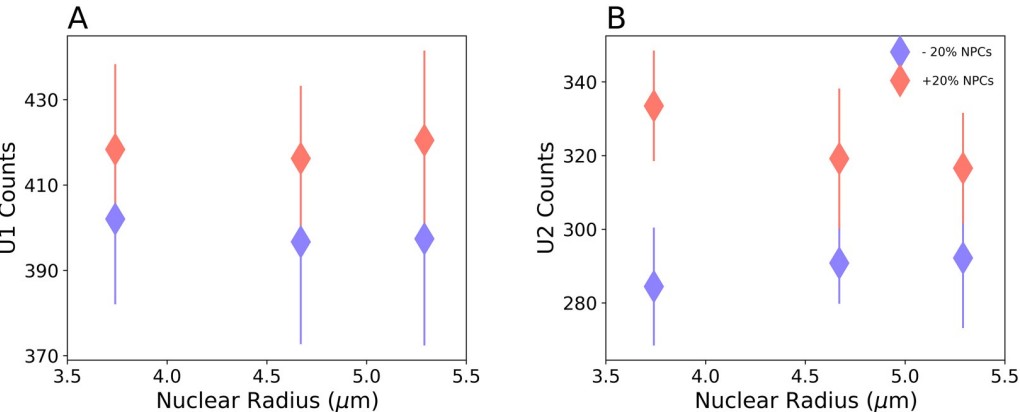

**Fig 3. Spliceosomal particle formation depends on NPC count.** Increase (pink) or decrease (blue) in the number NPCs by 20% results in a corresponding change in the number of U1 (A) and U2 (B) particles formed. The effect is consistent for different nuclear sizes. P-values for U1 results are: $1.8 \times 10^{-2}$ (3.7 $\mu$m), $6.8 \times 10^{-3}$ (4.7 $\mu$m), $4 \times 10^{-3}$ (5.3 $\mu$m); and for U2 results are: $3.9 \times 10^{-12}$ (3.7 $\mu$m), $6.4 \times 10^{-6}$ (4.7 $\mu$m), $8.5 \times 10^{-5}$ (5.3 $\mu$m). Error bars represent the standard deviations. For each condition, 20 simulation replicates for duration of 30 seconds were performed.

results in a decrease in the number of formed splicing particles as shown in S1 Fig. We also studied the effects of mitochondrial morphology on the production of splicing particles, by generating a cell with network-like mitochondria as shown in S2 Text and S2 Fig (see Methods section for details). By performing simulations using this geometry, we observed that the morphology of the mitochondria does not significantly affect the production of splicing particles. The results are presented in S4 and S5 Tables.

Human cells require about $10^5$ splicing particles [46]. With our computed particle production rate of $\approx 13.7\ s^{-1}$, it will take about 2 hours to generate the required abundance of splicing particles. This time scale is well within the lifetime of human cells, which further validates our mammalian cell model.

To obtain insight into the formation of splicing particles, we dissected the overall kinetics of the process in terms of discrete reactions occurring in each compartment, i.e., the nucleus and cytoplasm. These reactions include the transcription of snRNA and formation of (U1snRNA$_{nuc}$), cytoplasmic production of U1snRNA $\cdot$ Sm$^7$, and finally the assembly of mature U1. We determined the timescale for the formation of each of these three species within the first assembled U1 particle. The series of cytoplasmic reactions takes the longest to complete, irrespective of nuclear size, as shown in S3 Fig.

We examined the importance of multi-compartmentality by allowing all particle assembly steps to occur solely in the nucleus. As discussed above, in higher eukaryotes different components of the splicing particles join the assembly in different compartments [6]. This separation likely allows for higher quality control and prevents mixing of the partially-assembled particles with their substrates, thus preventing partially formed spliceosomes from deleteriously modifying pre-mRNAs. We postulated that assembly in the nucleus alone will result in snRNA binding to proteins in an incorrect order, or in incomplete assembly of the splicing particles. The series of uni-compartmental reactions can lead to the formation of misassembled splicing particle through reactions such as: U1snRNA $\cdot$ Sm$^5$ + U1$_{prot}$ $\rightarrow$ RNA $\cdot$ Sm$^5$ $\cdot$ U1$_{prot}$ reaction. The complex assembly of Sm-core on snRNA is followed by snRNA modification that triggers the nuclear import of the snRNA bound to Sm core [6]. Therefore, in the multi-compartmental assembly process, the misassembled splicing particle (RNA $\cdot$ Sm$^5$ $\cdot$ U1$_{prot}$ complex) is not found in the nucleus.

As an outcome of simulating the nuclear assembly of the splicing particles, we found that although the system can make fully assembled splicing particles, it produces significantly more misassembled particles (771 ± 25) as compared to mature particles (248 ± 15), since the former are not required to go through the full assembly cycle (see Fig 2A). This simulation result demonstrates the critical need for the compartmentalization of the overall assembly of the splicing particles. Similar multi-compartmental processes have been observed for other cellular machines such as ribosomal subunits [6].

## Nuclear speckles enhance effective splicing rate and control noise in mRNA

Nuclear speckles are self-organized liquid droplets that act as stores for splicing particles, and provide a mechanism to enhance other processes such as DNA repair and RNA modifications [5, 11]. Evidence suggests that a subset of splicing events occur within or at the periphery of these nuclear bodies [47] and speckles can have internal sub-organization [48]. Nuclear speckles, being liquid-liquid phase-separated regions, can promote certain biochemical reactions due to an enhanced concentration of the reactants [2]. To examine this phenomenon, we developed a reaction network to account for the effect of speckles on spliceosomal assembly and splicing (see Fig 2B). This network was included in stochastic RDME simulations containing speckles in the HeLa cell and we determined the resulting effect on mRNA production post-splicing and noise. The speckles were comprised of a concentrated store of splicing particles simulated as follows (see Methods section). We set the probability, $P_n$, for splicing particles to transit from the nucleus into the speckles higher than the probability, $P_s$, for the reverse transition. The higher the imbalance ($P_n/P_s$), the greater the degree of localization of splicing particles in the speckles (see Fig 4A). We compared mRNA production in cells with different degrees of splicing particle localization in the speckles with a control cell containing no speckles and splicing particles (U1) randomly distributed throughout the nucleus. Relative to the scenario containing no speckles (U1 fraction in speckles = 0 in Fig 4), a cell with about 10% of U1 located in speckles showed a large enhancement in the number of spliced mRNA transcripts from 0.25 to 60, which is effectively a $\approx$ 250-fold amplification (Fig 4B and 4C). Thus, even a slight increase in the localization of splicing particles in speckle, greatly enhances mRNA production. After the initial steep increase, the rate of mRNA production and enhancement, decrease and saturation is reached. This is because, as more particles are being localized in nuclear speckles, more pre-mRNA transcripts undergo splicing reactions, until almost all transcripts at a given time are bound to the splicing machinery (Fig 4B). This is verified by the reduction of average number of free transcripts from 53.3 to 0.2, for the cells with no speckles and with a localization of 85% U1 within speckles, respectively. Using green fluorescent protein labels, Rino et al. determined the ratio of splicing protein U2AF in speckles to that in the nucleus to be 1.27 ± 0.07 [49], which translates to 56% of U1 in speckles with respect to its total number. Strikingly, this experimentally determined ratio corresponds in our model to a $\sim$ 57% localization of splicing particles in speckles, which validates our model.

Nuclear speckles not only enhance splicing activity, but they also help limit the noise that splicing introduces into the whole gene expression process. To see this, we examined the effect of speckles on the noise associated with mRNA production, estimated in terms of the coefficient of variation (CV), which is the ratio of standard deviation of mRNA count to the average mRNA count. For CV calculations, we performed simulations for a fixed amount of time and the average and standard deviation were calculated from the value of mRNA at the end of the simulation time over 20 simulations. As the percentage of splicing particles in speckles increases (Fig 4D), the noise associated with spliced mRNA, decreases.

                    

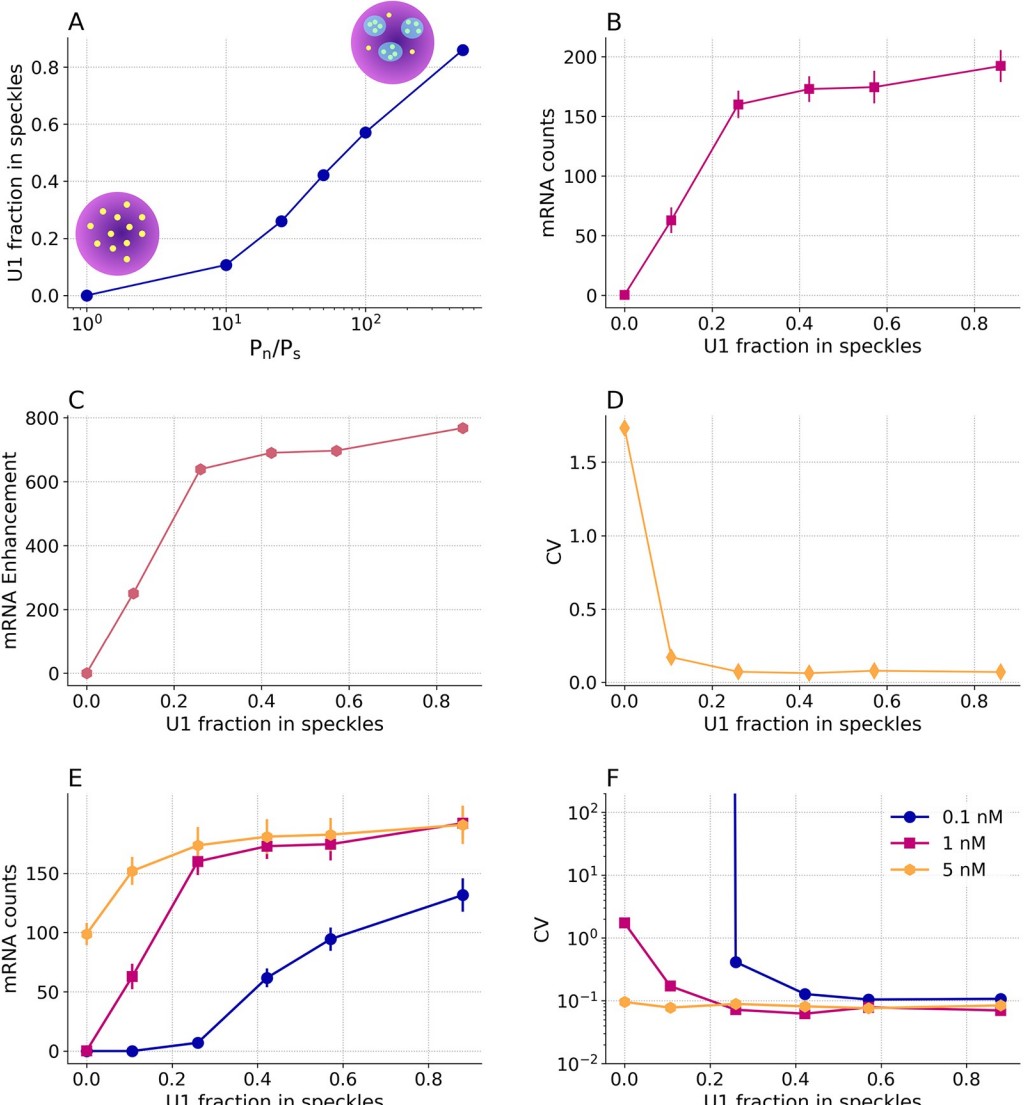

**Fig 4.** Splicing efficiency increases in the presence of speckles in the cells: A) The higher the probability for the splicing particles to transition from the cell nucleus to the speckles, relative to the reverse transition, the higher the localization of splicing particles in speckles. Schematically, the randomly distributed splicing particles (yellow dots) in the cell nucleus (colored in purple), localize in nucleus speckles (blue shaded regions) as the probability imbalance increases (the splicing particles concentration is 1 nM). B) As the percentage of splicing particles located in speckles increases, the number of spliced mRNA also increases. C) This enhancement in mRNA production is highly sensitive to the localization of splicing particles in speckles; with only a 10% localization of splicing particles in speckles, the splicing reaction is enhanced ∼ 250-fold relative to the case with no speckles. D) Noise estimated as coefficient of variation (CV), decreases as a greater percentage of splicing particles are localized in speckles. Splicing particle concentration affects the functional advantage of speckles: E) Enhancement in mRNA production due to the presence of speckles, depends on the U1 splicing particle concentration; F) Effect of the U1 splicing particle concentration on the mRNA production noise. Note that for cells with 0.1 nM U1, below 25% splicing particle localization, the CV is not defined due to lack of mRNA production. For each condition, 20 simulation replicates were performed. For simulation details see Methods.

**Speckle-enhanced splicing is concentration-dependent.** The number of splicing particles required per pre-mRNA transcript is a function of many variables including the rate of transcription [42] and therefore this number may vary from one gene to another. We investigated how variation in the ratio of splicing particles to pre-mRNA transcripts affects overall

mRNA transcript production and noise in a cell with nuclear speckles. Specifically, for 20 constitutively transcribing genes, we changed the number of particles available for pre-mRNA binding and splicing from 16 to 803 corresponding to a concentration range of 0.1-5 nM. The remainder of the total of $10^5$ splicing particles [46] was bound to pre-mRNA transcripts actively in the splicing process. Fig 4E summarizes how the concentration of U1 splicing particles affects the ability of speckles to enhance splicing. For cells with 0.1 nM U1, the absence of speckles results in no generation of mRNA. In fact the no mRNA production was observed until the U1 localization reaches 25%. At 1 nM U1, mRNA production in a cell with speckles is $\approx$ 770-fold that for a cell with no speckles. On the contrary, at 5 nM, there is little enhancement. Thus, at lower concentrations of U1, speckles enhance splicing much more strongly. Consistently, as Fig 4F shows, the noise of mRNA production is also influenced. Because of the lack of mRNA production in control cells with no speckles, the CV for cells with 0.1 nM U1 is not defined. At 1 nM U1, the noise in the presence of speckles is $\approx$ 25-fold lower as compared to a cell with no speckles; whereas, at 5 nM, the noise is unaffected by whether speckles are present or not.

## Speckle size and number have been fine-tuned by cells to optimize mRNA production

A physical constraint of phase separation is that the the concentration of splicing particles within speckles remains roughly constant. Thus we investigated how a cell controls the number and the size of nuclear speckles. We hypothesize that the experimentally observed anatomy of the speckles is optimized by the environment inside cells. To test this hypothesis, we assigned about 10% of the nuclear volume to speckles [18]. Keeping the total volume of speckles constant, we increased the number of speckles and reduced their sizes, as shown in Fig 5A. Increasing the number of speckles results in increasing their surface area (Fig 5A), which in turn enhances the pre-mRNA splicing, as shown in Fig 5B. This is because the higher surface area increases the probability of splicing particles to diffuse into the nuclear speckles resulting in increased localization. However, beyond $\sim$ 50 speckles the number of produced mRNA plateaus, due to the compensation of splicing particle localization by relatively smaller-sized nuclear speckles. Production of mRNA was maximized when there were between 20 and 50 speckles, which coincides directly with the experimentally determined values [11]. In addition,

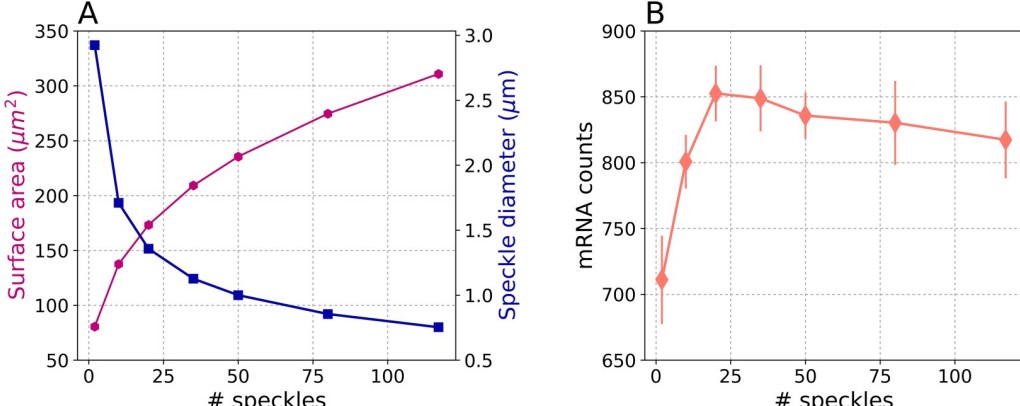

**Fig 5. A) Increasing the number of nuclear speckles, results in an increase of the surface area (magenta curve) and decrease of the speckles diameter (blue curve); B) mRNA production increases as the number of speckles increases till about 50 speckles beyond which the production plateaus.** Error bars represent the standard deviations. For each condition, 20 simulation replicates were performed.

the size of the nuclear speckles corresponding to the maximum mRNA production falls between 1.4 to 1 $\mu$m, which is also compatible to the known nuclear speckles diameters of one to a few microns [11]. Therefore, our results suggest that it is plausible that the cells optimize the design parameters of speckles (the number and size) to maximize the mRNA production.

## Gene distribution around speckles affect transcript splicing and mRNA production

It is known that genes are organized non-randomly around nuclear speckles [32, 33]. In a recent study, Chen et al. investigated the organization of whole genome using TSA-seq method [34]. They showed that the most highly expressed genes are located between $\approx 0.05$ and 0.4 $\mu$m from the periphery of a speckle. It was also speculated that the genome movement of several hundreds of nanometers from nuclear periphery towards speckles could have functional significance. To test their hypothesis, we investigated the effects of active genes distribution around speckles' peripheries. We varied the gene distance from 0.05 to 0.94 $\mu$m and observed the effect on the number of spliced mRNA transcripts found in cytoplasm, but without explicitly accounting for transcriptional machinery. As Fig 6 demonstrates, increasing the distance of the genes to speckle periphery from $\approx 0.05$ to 0.1 $\mu$m sharply decreases the mRNA counts by a factor of 2, with no further significant decrease at larger distances. Thus, the speckle-gene proximity effect might be even more pronounced over a short distance range than Chen et al. were able to resolve.

Since our model does not involve movement of speckles toward an active transcription site, the observed effect is mainly due to the diffusion of the transcripts in the nucleus before they

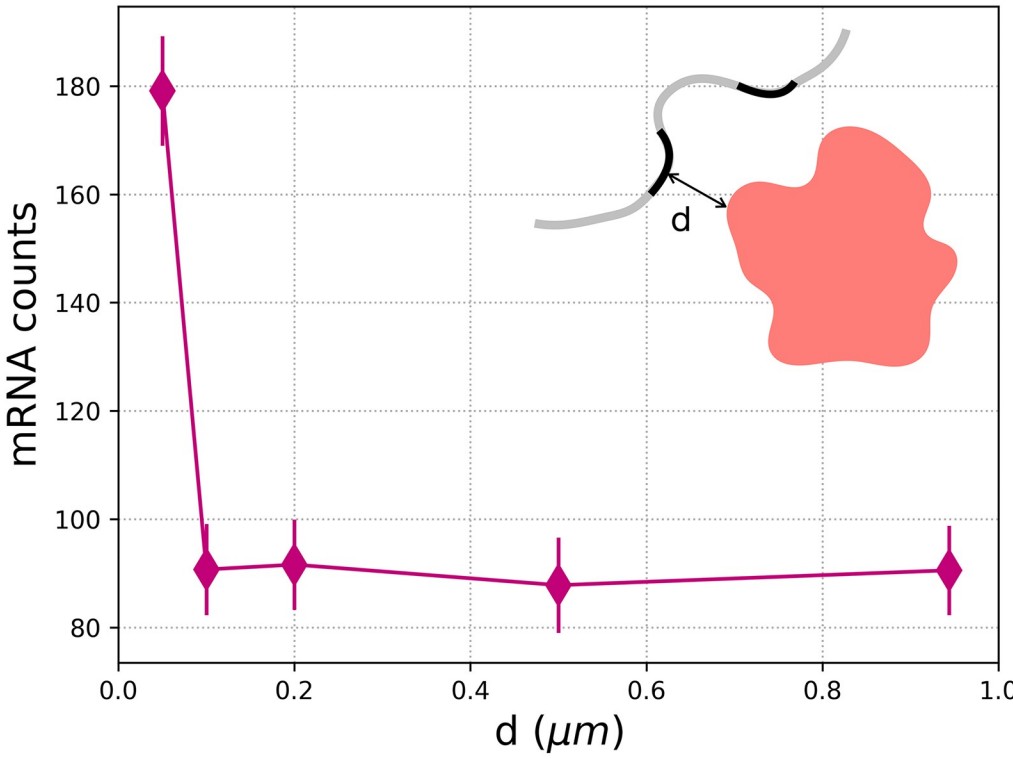

**Fig 6. Our model predicts that the mRNA production decreases by a factor of two, above 0.05 $\mu$m.** An schematics shows a nuclear speckle in red and the distance of active genes on chromatin (curved line in grey) from the speckle periphery (d). Error bars represent the standard deviations. For each condition, 20 simulation replicates were performed.

become associated with the speckles. Therefore, our model makes the simple prediction that the gene distribution around speckles has an effect on mRNA splicing. It is plausible that this effect might be regulated by speckle movement towards transcriptionally active genes, consistent with the fluid nature of these nuclear bodies.

## Discussion

Spatial organization is one of the key features of eukaryotic cells that brings order to complex biochemical reactions. We studied two aspects of this organization both in connection with RNA splicing (the process of removal of non-coding regions from pre-mRNA transcripts). We investigate, firstly, the effects of variation of cellular organelles on assembly of the building blocks of splicing; and secondly, mRNA production at nuclear speckles which are membrane-less organelles that have higher concentrations of reactants involved in splicing. We constructed a spatially-resolved mammalian cell (HeLa cell) model from a variety of experimental data. Using this model, we simulate the splicing event with a kinetic model we developed, as stochastic reaction-diffusion processes.

We show that number of nuclear pore complexes—assemblies of proteins embedded in the nuclear envelope that control the traffic between nucleus and cytoplasm—affects the splicing particles formation rate for different nuclear sizes, and the particles formation is more efficient when the precursors are produced in different compartments before maturation. The localization of splicing particles in nuclear speckles enhances the mRNA production and reduces its associated noise. We suggest a rationale for the number and size of speckles based on optimal resulting mRNA concentrations. By demonstrating the effects of the genes distribution around our speckles on mRNA counts, we propose that the movement of nuclear speckles toward active chromatin regions could be regulated by cells to control the transcripts production.

### Limitations of the current spatially-resolved human cell model

While novel, our model approximates the underlying biophysics, and therefore has some limitations:

1. There is a lack of experimental data for describing the rates of individual reactions within the splicing process. To overcome this challenge, we defined approximate lumped reactions (e.g., NPC transport and Sm proteins binding to snRNA [39]) and assigned rates based on either available experimental data [50] or simple models, such as diffusion-limited reactions. A potential alternative to solving this problem is detailed in the next paragraph.

2. Our current model includes nucleus, nuclear speckles and Cajal bodies within it, and cytosolic organelles such as ER, Golgi apparatus, and mitochondria, that are near the nuclear periphery and contribute substantially to the excluded volume effects. However, it does not yet include compartments or structural features such as chromatin, the nucleolus, microtubules, actin filaments, endosomes, peroxisomes, and lysosomes. For the purposes of modeling RNA splicing process, none of these compartments (aside from chromatin) are thought to play a significant role. For a complete spatial model of the human cell, especially when studying processes that are involved with these organelles/compartments, including them will be required.

3. For simplicity we considered a constitutively spliced gene in our studies. To account for complexity within human cells, alternative splicing should be also included which itself requires the inclusion of splice-sites selection. Splice-sites selection is affected by inter- and intracellular signaling pathways. [51].

### Challenges of simulating the complete cell cycle of the entire human cell and strategies for overcoming them

There are many challenges to create a dynamical model for the entire human cell. Here, we provide three challenges that in our opinion are the most immediate ones:

1. For simulating the complete life cycle of a eukaryotic cell two additional components will be required: a) inclusion of metabolism, and a more complete description of the genetic information processes (transcription, translation, and DNA replication). Although our code presents a platform for the inclusion of these processes, however, as demonstrated by Karr et al. [52], even for bacterial systems, accounting for these processes will add considerable complexity to the model –especially as they are distributed among different compartments in eukaryotic cells; b) addition of the remaining organelles such as microtubules, actin filaments and the nucleolus. We envision they can be constructed following a similar approach as outlined in detail in Methods. For representation of sub-compartments such as chromatin, rich sets of experimental data for instance from the 4D nucleome project [53] are becoming available. Such developments will enable modelers to improve the existing model and to capture chromatin dynamics [54, 55]. We hope that extension of the model will be a community process and provide other researchers with the tools to make it so.

2. Modeling a comprehensive set of cellular processes requires an extensive amount of computational time, even when performing the simulations on multiple GPUs. Such studies can be accelerated by employing hybrid methodologies. As an example, parts of the cellular processes involving a high concentration of species (e.g., metabolites) can be simulated as ordinary differential equations (ODEs) that can be integrated with stochastic simulations such as chemical master equations (CMEs) for treating the processes like transcription. Recently, a hybrid CME-ODE method was implemented in the LM and can provide a 10-50-fold speedup in comparison with pure CME simulations [56].

3. For describing an extensive set of biological reactions, especially in eukaryotic cells, the rates of individual reaction steps are often not yet available from experimental data. To overcome this barrier, two approaches can be taken: a) to estimate/infer the rates from computational simulations of each specific reaction using methods such as enhanced sampling techniques [57, 58] and quantum mechanical calculations [59]; however, these simulations are computationally expensive and time-consuming; and b) to introduce lumped reactions, based on available experimental data describing the biological processes and time-scale analysis [50] and c) Bayesian approaches using a multitude of experimental data elements to fit unknown rates.

### Model extensibility

To build upon our model, we provided a documentation within our Python package which describes how to add/ remove reactions; modify the number, size and morphology of the organelles currently present in the model; and add new organelles. The code can be edited within the widely-used Jupyter notebooks. As discussed in the Introduction, to perform hour-long simulations of the entire human cell, using GPU acceleration is an inevitable architecture choice. Because our LM software excels on GPU computing, and such technologies are currently available to the majority of researchers, our platform is particularly primed for extensions to various cell types and longer simulations. The Python code for setting up a HeLa cell model is available at: https://eukaryoticcellbuilder.github.io/HeLa_Builder/.

Overall, our reaction-diffusion model of spliceosome assembly and function in a realistic mammalian cell environment that included some of the most relevant constituents, allowed us to meaningfully connect the cellular geometry to the underlying biological processes. At the same time, we expect the presented cell model to provide a versatile platform for studying processes with spatio-temporal elements beyond mRNA splicing. Having said that, our platform presents one of the many steps required to be able to seamlessly integrate various models and fully simulate the complexity of a eukaryotic cell across different length and time scales. Such an integration can eventually lead to the prediction of inter- and intra-cellular emergent phenomena.

## Methods

### Construction of a representative spatially-resolved HeLa cell model

Spliceosomal assembly and activity consists of multi-compartmental, reaction-diffusion processes, necessitating a spatial representation of the cellular geometry. HeLa cells are an optimal model system as they have been the subject of extensive investigations exploring cell geometry and cellular composition. Additionally, data from individual measurements of specific components (e.g., size, morphology, relative mass fraction) were used to inform the construction of a representative cell model [11–18, 24–27]. A constructive solid geometry (CSG) approach, wherein basic geometric objects are combined programmatically via set operations (e.g. unions, differences, intersections), was used to build the HeLa cell. Since LM (v 2.3) [19, 20] requires that each location within the space be defined as a single site-type the various CSG objects were stenciled onto the simulation lattice in "depth order" (also called the "Painter's algorithm"). Overall, our model consists of 11 different site-types including: 1) extracellular space, 2) plasma membrane, 3) cytoplasm, 4) nuclear membrane, 5) nucleoplasm, 6) Cajal bodies, 7) nuclear speckles, 8) nuclear pore complexes, 9) mitochondria, 10) Golgi apparatus and 11) endoplasmic reticulum (ER). The overall simulation volume was constructed as a cubic box with 18.432 $\mu$m side-length. The space was discretized into a cubic lattice of points spaced 64 nm apart. HeLa cell volumes have been measured at 2600, 3000 and 4400–5000 $\mu m^3$ [25, 60, 61]; we chose the mid- size cell as a template for our model. HeLa cells that are grown in suspension appear spherically-shaped, so we chose to design the overall cell architecture as a sphere with radius 8.9 $\mu$m. Nuclei have measured volumes of, 220 and 374 $\mu m^3$ [26, 27]. Refs. [25] and [13] suggested nuclear volumes corresponding to 10% and 21.1% of the total cell volume. As we wanted to test the importance of nuclei size on splicing, multiple nuclear radii were investigated, including 3.7, 4.2, 4.7 and 5.3 $\mu$m corresponding to all the above-motioned volumes and volume-fractions. Plasma and nuclear membranes were implemented as a thin sheet of lattice points (128nm thick) separating the extracellular space, the cytoplasm and the nucleus. The Golgi apparatus was constructed as an intersection of a cone with several spherical shells of various radii placed successively from the edge of the nucleus into the cytoplasm. The apex of the cone was centered in the cell with the based positioned deep in the cytoplasm. In this way, the Golgi roughly approximates what is seen in experiments. Nuclear speckles and Cajal bodies were modeled as spheres placed randomly within the nucleus. Mitochondria were modeled as randomly oriented spherocylinders placed within the cytoplasm. A network-like mitochondria was also constructed by randomly placing 2.95 $\mu$m-long rods that can cross each other, while keeping the total volume of the mitochondria constant, as shown in S2 Fig. Nuclear pore complexes (NPCs) were embedded in the nuclear envelope. NPCs were constructed as a set of spheres of radii 0.08 $\mu$m to ensure connectivity from the nucleus to the cytoplasm. Sizes for these organelles can be found in Table 1. The ER was also constructed in a randomized fashion with the details and construction algorithm presented in Supplementary Information. Total counts of the organelles were based on either direct experimental

quantification or based on relative volume fraction measured for the overall cell. The endosomes, lysosomes, actin-cytoskeleton, peroxisomes in the cytoplasm; nucleolus and chromatin have not been included into the present version of the model. According to Ref. [13], each of the cytoplasmic organelles contribute less than 1% of to the total cell composition, and therefore, were not modelled. The nuclear components were chosen mainly among those that play a role in RNA splicing processes. A representative HeLa cell geometry resulting from this procedure is shown in Fig 1. The associated code for setting up a HeLa cell model for LM [19, 20] is available at: https://eukaryoticcellbuilder.github.io/HeLa_Builder/.

## Kinetic models

### Implementation of nuclear speckles

Nuclear speckles were modelled as spherical regions in the nucleus with radii of 0.35 $\mu$m. Previously, the splicing particles localization in speckles has been implemented by assuming a higher affinity for splicing particles to bind to unknown binding partners in speckles with respect to binding to pre-mRNA transcripts in the nucleoplasm [49]. We imposed an imbalance on the transition probabilities for the splicing particles and pre-mRNA transcripts to move from the nuclear speckles to the nucleoplasm, and vice-versa. This approach, as shown in the Results section, will reproduce experimentally observed concentration ratio of splicing particles between the speckles and the nucleoplasm [54], additionally, the presence of dummy particles in the speckles are not required. Specifically, the probability of the splicing particles to move from the nucleoplasm regions to the speckles ($P_n$) was higher than the reverse direction ($P_s$). To examine the effect of the bias, the $P_n/P_s$ values were varied. With increasing $P_n$ values, more particles accumulate in the speckles and the nucleus becomes more diluted. Fig 7 shows the localization of splicing particles in speckles upon application of this bias in our model.

### RDME simulations methodology in LM

We use formalism known as the reaction-diffusion master equation (RDME), which describes the state of a stochastic system to be in a specified spatial and compositional "state", as the

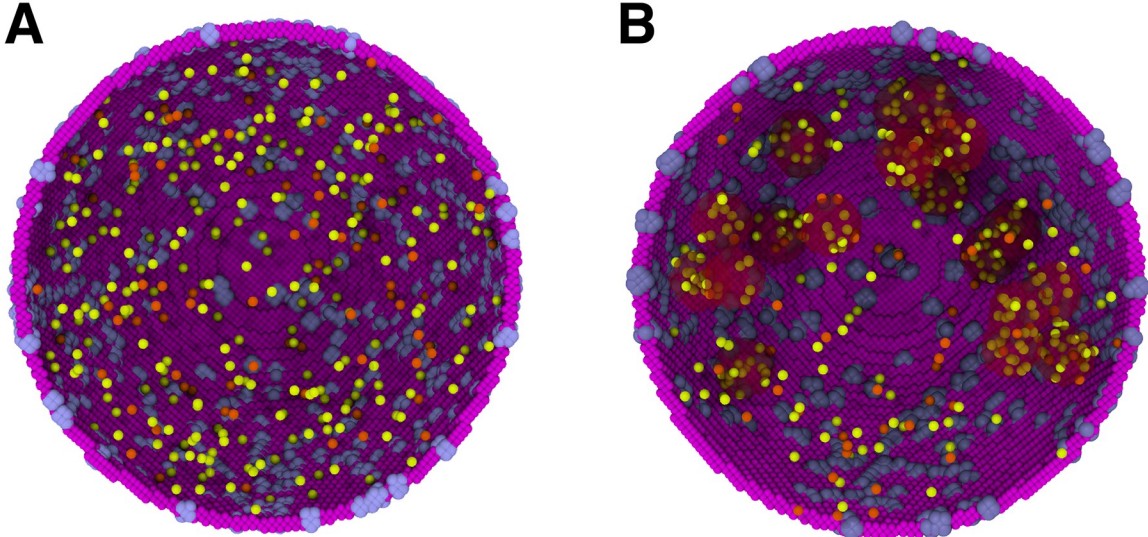

**Fig 7. Formation of speckles in our simulations: A) splicing particles (U1 and U2 colored in yellow and orange, respectively) diffusing freely in the nucleus without speckle, B) Introduction of an imbalance on transition probabilities of splicing particles from the nucleus to speckles results in the localization of the splicing particles in the speckles shown as red-shaded regions.**

 

underpinning for the simulations in this paper. For systems of our interests, this state comprises of the number of each interacting species along with their locations. In RDME simulations using LM, the geometry of the system is first discritized to a cubic grid with a lattice spacing $\lambda$. Additionally, a fundamental time step $\tau$ is specified. The time evolution of the probability of the system to be in specific state $\mathbf{x}$, containing number of molecules for each of the $N_{sp}$ species present at each subvolume $v \in V$, is determined by the rates of change due to reaction and diffusion, defined by operators $\mathbf{R}$ and $\mathbf{D}$, respectively:

$$\frac{dP(\mathbf{x}, t)}{dt} = \mathbf{R}P(\mathbf{x}, t) + \mathbf{D}P(\mathbf{x}, t),$$

$$\frac{dP(\mathbf{x}, t)}{dt} = \sum_{v}^{V}\sum_{r=1}^{N_{rxn}}[-a_r(\mathbf{x}_v)P(\mathbf{x}_v, t) + a_r(\mathbf{x}_v - \mathbf{S}_r)P(\mathbf{x}_v - \mathbf{S}_r, t)]$$

$$+ \sum_{v}^{V}\sum_{\zeta}^{\pm \hat{i}, \hat{j}, \hat{k}}\sum_{\alpha=1}^{N_{sp}}[-d_v^{\alpha}x_v^{\alpha}P(\mathbf{x}, t) + d_{v+\zeta}^{\alpha}(x_{v+\zeta}^{\alpha} + 1)P(\mathbf{x} + \mathbf{1}_{v+\zeta}^{\alpha} - \mathbf{1}_v^{\alpha}, t)]. \quad (1)$$

The first two terms of this equation define the probability flux within each subvolume due to reactions: $a_r(\mathbf{x}_v)$ is the reaction propensity for reaction $r$ of $N_{rxn}$ reactions and $\mathbf{S}_r$ is the $N \times R$ stoichiometric matrix describing the net change in molecules number when a reaction occurs. The last two terms describe the rate of the change of the probability due to molecules' propensity to diffuse between neighboring subvolumes. $d_v^{\alpha}$ is the diffusive propensity for a molecule of species $\alpha$ to jump to a neighboring subvolume, which is related to the macroscopic diffusion coefficient by $d_v^{\alpha} = \frac{D_v^{\alpha}}{\lambda^2}$. The first part of the diffusion operator defines the probability flux out of the current state due to the diffusion of molecules from subvolume $v$; and the second part of the diffusion operator, describes the diffusion of molecules into the current state from subvolumes, $v + \zeta$, where $\zeta$ defines the neighboring subvolume in $\pm x$, $\pm y$, $\pm z$ directions.

Using an approximation for sampling Eq 1, as implemented in LM, simulations of biologically relevant timescales ($\sim 15$ min) for the HeLa cell model on multiple GPU hardware became possible [19, 20].

## RDME simulation details

**Formation of splicing particles.** In addition to structural features discussed above, the abundance of the proteins participating in the formation of splicing particles were derived from proteomics data of HeLa cell [17]. The concentration of cytoplasmic proteins of $G^5$, $Sm^5$ and $Sm^2$ was 0.61 $\mu M$; the nuclear proteins of $U1_{prot}$ and $U2_{prot}$ were 0.89 $\mu M$ and 0.56 $\mu M$, respectively. Moreover, the number of active snRNA genes have been determined to be 30 in haploid cells [66]. These abundances were used as the initial condition for the simulations. Proteins and RNA molecules were randomly distributed throughout their designated regions with locations sampled from a uniform distribution; for instance, cytoplasmic proteins ($G^5$, $Sm^5$ and $Sm^2$) were distributed throughout the cytoplasm, at sites which were not occupied by mitochondria or ER. For each separate simulation replicate, the placement of proteins and active genes, together with cellular organelles such as: mitochondria, ER, nuclear speckles, NPCs and Cajal bodies, were randomized. The RDME simulations were performed for 30 s of biological time eight Tesla GPUs with a walltime of 30 minutes. The time step was $7.3 \times 10^{-5}$s and the lattice spacing was 64 nm.

**pre-mRNA splicing within nuclear speckles and nucleoplasm.** For studying the RNA splicing, as discussed in the main text, we started with already assembled splicing particles (U1 to U6) and distributed them randomly in the nucleoplasm. 20 constituitively transcribing active genes were localized around 20 randomly placed nuclear speckles. These simulations

were performed for 15 minutes of biological time on eight Tesla K80 GPUs with a walltime of 20 minutes. The simulation details were: time step of $3.3 \times 10^{-3}$s, lattice spacing of 64 nm, nucleus size of 4.2 $\mu m$ with 1515 NPCs.

## Supporting information

**S1 Text. Endoplasmic reticulum construction.**
(TEX)

**S2 Text. Effects of mitochondria network-like morphology on the splicing particles formation.**
(TEX)

**S1 Fig. Changes in the number of NPCs, correlates withe the number of formed U1 splicing particles.**
(PNG)

**S2 Fig. The HeLa cell with a network-like mitochondria.** See Methods section for construction details of the mitochondria.
(PNG)

**S3 Fig. The time required to form the first splicing particle dissected by the set of discrete reactions occurring in the nucleus and cytoplasm.** Error bars represent the standard deviations. For each condition, 20 simulation replicates were performed.
(PNG)

**S1 Table. Parameters used in the cellular automata program to create realistic ER.**
(TEX)

**S2 Table. Diffusion coefficients for species involved in splicing particles formation.** Abbreviations are: nucleus (N), NPC (P), Cajal bodies (J), nuclear speckles (S) and cytoplasm (C).
(TEX)

**S3 Table. Diffusion coefficients of spliceosomal particles.** Abbreviations are: nucleus (N) and nuclear speckles (S).
(TEX)

**S4 Table. The results of the simulations sets with a network-like morphology of the mitochondria as compared to the fragmented-mitochondria that were reported in the main text.**
(TEX)

**S5 Table. The p-values of data from various simulation sets.**
(TEX)

## Acknowledgments

Z.G. thanks Prof. Andrew Belmont and Dr. Joseph Dopie (MCB, U of Illinois) for many stimulating conversations, and Prof. Andrew Belmont, Prof. Prashant Jain (Chemistry, U of Illinois), Prof. K. Prasanth (MCB, U of Illinois), Dr. Marian Breuer (Chemistry, U of Illinois) and Dr. Pankaj Chaturvedi (MCB, U of Illinois) for critical reading of the manuscript and helpful comments. Z.G. especially thanks Mike Hallock (School of Chemical Sciences, U of Illinois) for support with LM. Supercomputer time was provided by XStream-XSEDE [grant TG-MCA03S027].

## Author Contributions

**Conceptualization:** Zhaleh Ghaemi, Joseph R. Peterson, Martin Gruebele, Zaida Luthey-Schulten.

**Data curation:** Zhaleh Ghaemi.

**Formal analysis:** Zhaleh Ghaemi.

**Funding acquisition:** Martin Gruebele, Zaida Luthey-Schulten.

**Investigation:** Zhaleh Ghaemi.

**Methodology:** Zhaleh Ghaemi, Joseph R. Peterson.

**Software:** Zhaleh Ghaemi, Joseph R. Peterson.

**Supervision:** Martin Gruebele, Zaida Luthey-Schulten.

**Validation:** Zhaleh Ghaemi.

**Visualization:** Zhaleh Ghaemi.

**Writing – original draft:** Zhaleh Ghaemi, Joseph R. Peterson, Martin Gruebele, Zaida Luthey-Schulten.

**Writing – review & editing:** Zhaleh Ghaemi, Joseph R. Peterson, Martin Gruebele, Zaida Luthey-Schulten.

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
