## [Decision Letter · Decision Letter 0]

7 Oct 2019

Dear Dr Ghaemi,

Thank you very much for submitting your manuscript 'An In-Silico Human Cell Model Reveals the Influence of Spatial Organization on RNA Splicing' for review by PLOS Computational Biology. Your manuscript has been fully evaluated by the PLOS Computational Biology editorial team and in this case also by independent peer reviewers. The reviewers appreciated the attention to an important problem, but raised some substantial concerns about the manuscript as it currently stands. While your manuscript cannot be accepted in its present form, we are willing to consider a revised version in which the issues raised by the reviewers have been adequately addressed. We cannot, of course, promise publication at that time.

Specifically, in the revised version please clarify which features of the whole cell model are truly needed to reach the conclusions of the study, since as it seems, most of the claimed data integration into a whole cell model is not used. A further point is that it is stated the software"will be available". We need a more clear statement on this and a specific link where the software can be reached.

Sincerely,

Attila Csikász-Nagy

Associate Editor

PLOS Computational Biology

Jason Haugh

Deputy Editor

PLOS Computational Biology

[LINK]

Reviewer's Responses to Questions

**Comments to the Authors:**

Reviewer #1: Whole-cell models that connect molecular biology with phenotype are needed to help researchers integrate disparate data, assemble a comprehensive understanding of biology, and enable experiments that are inaccessible to conventional experimental methods. Toward whole-cell models, Ghaemi et al. report a spatially-detailed detailed computational model of splicing in HeLa cells and its usage to elucidate several aspects of the role of spatial organization in splicing. The study is innovative for developing one of the first mechanistic models of splicing -- an essential aspect of the biochemical complexity of eukaryotic cells, for reporting some of the longest (15 minutes) spatially detailed simulations of entire eukaryotic cells, and for several novel quantitative analyses of splicing. These innovations are important, especially given the substantial challenges to building and simulating models of whole-cells. The methods are thorough and the manuscript is well-written. However, we have several minor comments about several aspects of the manuscript which could be clearer. In addition, we think that the manuscript would be strengthened by an expanded conclusion about the limitations of the reported model and the challenges going forward to building and simulating models that capture more chemical complexity, more intracellular processes, more molecules and spatial compartments, and longer lengths of time.

Minor comments

- Because the reported model represents a portion of a HeLa cell and there are multiple complementary lines of whole-cell modeling research ongoing in the literature, we think would it be helpful for readers to clarify what is meant by a HeLa "whole-cell model" on pp 5. At the authors discretion, this could be described in an additional paragraph in the introduction or conclusion. In particular, it would be helpful/interesting to discuss what's missing from the model and what's needed to capture whole cells. For example, what is needed to capture splicing site selection, the combinatorial number of transcripts that could be created from each gene, how to handle the complexity of simulating the entire genome, etc.?

- (optional) In the interest of encouraging additional whole-cell modeling research, we think it would be helpful to comment on the extensibility of the model/methods at the end of the conclusions. While the reported model is an advance, realistically it would likely be difficult for most researchers to build upon the implementation, given lack of easy to use whole-cell modeling tools. It would be helpful to comment on how extensible the present model is and what technology is needed to make models like this easier to build and extend.

- Please clarify several methodological details:

- pp 5: "target volume fractions": Is this intended to mean the modeled volumes of the organelles?

- pp 5: "The [essential] components of the cell include: ...": Is this intended to mean essential for the splicing processes that will be investigated?

- pp 5: "... genes ... were placed around the speckles ...": Please clarify how they were placed.

- pp 7: "... assembles in a [stepwise] manner ..."

- pp 7: "... pre-mRNA is [converted] to an mRNA transcript ..."

- pp 8: "... splice site has been chosen ...": Which sites does the model choose?

- pp 10: "... production rate of ...": Does this account for the turnover of the particles?

- Lightening the background in Figure 2A might make the reactions easier to read.

- For consistency, should "Complex A" in Figure 2B show one rather than two U1 particles?

- The word choice and grammar could be clarified throughout the manuscript. Below are several examples:

- pp 3: "without massively increasing [the] gene count ..."

- pp 3: "The [order of] intronic removal ...". Is this intended to mean temporal order?

- pp 3: "... coding regions can be shuffled after the removal ...". Do you intend to mean "shuffle" as in exon scrambling? The statement also seems to confuse the order of events; the coding sequence is determined by splicing.

- pp 3: "... occurs in multiple steps ..., [transport between ] .., and terminates ...". We think transport should be plural

- pp 5: The inconsistent use of past and present tense in reference to the distribution of units makes it unclear whether the statements refer to the biology or to the model. Consistent tense would clarify that these statements refer to the model.

- pp 7: Correct comma placement in "... reach the cytoplasm where, by a series of complex reactions, they bind ..."

- pp 8 and others: The first double quote in each pair of quotes should point to the right (e.g., "complex E", "complex B")

- pp 15: Remove the space in "nano meter" and pluralize

Reviewer #2: The manuscript by Ghaemi et al describes two main things. First, is a claimed “whole cell” model of a HeLa (mammalian) cell. It would seem to be a whole cell model in the sense that it is spatially resolved, being parameterized in a careful and detailed way from a variety of image based and biochemical composition data. However, the number of biochemical processes considered seems to be quite few to support the claim of a whole cell model (a few RNA splicing reactions). Second, they incorporate in their model a few reactions that describe, mostly in terms of lumped kinetic processes, pre-mRNA splicing. They use spatial kinetic master equation approaches to simulate reaction diffusion mechanisms underlying spliceosome formation, coalescence into nuclear speckles, and production of mature mRNA from pre-mRNA. While splicing is surely a central and important cellular process, the specific questions and rationale for their modeling here was not laid out clearly. Most of the findings from analyzing the model incorporating these processes seem fairly obvious, such as increasing the number of nuclear pores allows more spliceosomes to be formed because of increased nucleocytoplasmic transport, or that when pre-mRNA are localized with the splicing factors, splicing rates increase significantly. Moreover, the authors do not seem to use most of the spatial model features incorporated (besides nucleus vs. cytoplasm and nuclear subcompartments such as speckles). Overall, there does not seem to be significant advances or new knowledge presented in the manuscript to warrant publication in its current form. Some more specific points are listed below in addition.

-It seems that this model is of a spherical cell, when most HeLa lines are adherent (some are floating).

-Mitochondria form large networks, and are usually not thousands of individual compartments (as is currently modeled).

-Liquid liquid phase separation is known to be caused by intrinsically disordered proteins and/or multi-valent interactions, but no such mechanisms were used to invoke nuclear speckles, which were claimed to be separate liquid phases.

**Have all data underlying the figures and results presented in the manuscript been provided?**

Reviewer #1: Yes

Reviewer #2: Yes

PLOS authors have the option to publish the peer review history of their article (what does this mean?). If published, this will include your full peer review and any attached files.

Reviewer #1: Yes: Jonathan Karr and Bilal Shaikh

Reviewer #2: No

---

## [Decision Letter · Decision Letter 1]

8 Jan 2020

Dear Dr Ghaemi,

Thank you very much for submitting your manuscript, 'An In-Silico Human Cell Model Reveals the Influence of Spatial Organization on RNA Splicing', to PLOS Computational Biology. As with all papers submitted to the journal, yours was fully evaluated by the PLOS Computational Biology editorial team, and in this case, by independent peer reviewers. The reviewers appreciated the attention to an important topic but identified some aspects of the manuscript that should be improved.

We would therefore like to ask you to modify the manuscript according to the review recommendations before we can consider your manuscript for acceptance. Your revisions should address the specific points made by each reviewer and we encourage you to respond to particular issues Please note while forming your response, if your article is accepted, you may have the opportunity to make the peer review history publicly available. The record will include editor decision letters (with reviews) and your responses to reviewer comments. If eligible, we will contact you to opt in or out.raised.

- Supporting Information uploaded as separate files, titled 'Dataset', 'Figure', 'Table', 'Text', 'Protocol', 'Audio', or 'Video'.

We hope to receive your revised manuscript within the next 30 days. If you anticipate any delay in its return, we ask that you let us know the expected resubmission date by email at ploscompbiol@plos.org.

Sincerely,

Attila Csikász-Nagy

Associate Editor

PLOS Computational Biology

Jason Haugh

Deputy Editor

PLOS Computational Biology

[LINK]

Reviewer's Responses to Questions

**Comments to the Authors:**

Reviewer #1: The authors have addressed our major concerns. However, we feel that the manuscript would be strengthened by clarifying the purpose and goals of the study in the abstract, author summary, and introduction. In addition, although the HelaBuilder tutorial is easy to follow and the model is extensible *in principle*, in practice the procedural definition of the model and the lack of annotation within the model code will make it challenging for others to expand upon the model. In the interest of transparency, we think its important to acknowledge that much more needs to be done to make this model and other really extensible.

Minor comments

- knwon is mispelled on page 3

Reviewer #2: The authors have substantially revised the manuscript for clarity which has much improved it. The specific application to splicing remains somewhat lacking in this reviewer's opinion (mainly in terms of new biological knowledge). However, the bigger picture aspects of the manuscript, such as a spatially resolved mammalian cell model scaffold with available code for extension, and pushing the boundaries of simulation time for such a model, are appreciated and should benefit the literature and community.

**Have all data underlying the figures and results presented in the manuscript been provided?**

Reviewer #1: Yes

Reviewer #2: Yes

PLOS authors have the option to publish the peer review history of their article (what does this mean?). If published, this will include your full peer review and any attached files.

Reviewer #1: Yes: Jonathan Karr

Reviewer #2: No

---

## [Editor Report · Decision Letter 2]

6 Feb 2020

Dear Dr. Ghaemi,

We are pleased to inform you that your manuscript 'An In-Silico Human Cell Model Reveals the Influence of Spatial Organization on RNA Splicing' has been provisionally accepted for publication in PLOS Computational Biology.

Before your manuscript can be formally accepted you will need to complete some formatting changes, which you will receive in a follow up email. A member of our team will be in touch within two working days with a set of requests.

Best regards,

Attila Csikász-Nagy

Associate Editor

PLOS Computational Biology

Jason Haugh

Deputy Editor

PLOS Computational Biology

---

## [Editor Report · Acceptance letter]

2 Mar 2020

PCOMPBIOL-D-19-01350R2 

An In-Silico Human Cell Model Reveals the Influence of Spatial Organization on RNA Splicing

Dear Dr Ghaemi,

I am pleased to inform you that your manuscript has been formally accepted for publication in PLOS Computational Biology. Your manuscript is now with our production department and you will be notified of the publication date in due course.

With kind regards,

Laura Mallard
